# COMPUTER METHODS FOR GENOME TEXT COMPLEXITY AND ENTROPY ESTIMATES FOR VIRUSES AND BACTERIA

The development of high-performance DNA sequencing technologies serves as the basis for the growth of the volume of accumulated genomic data for an increasingly wide range of organisms – plants, bacteria and viruses, and an important resource for Artificial Intelligence (AI) systems. Computer analysis of genomes as text allows us to obtain new information, identify patterns and interdependencies between symbols for genome classifying, their evolutionary origin and potential pathogenicity. Development of the databases such as Ensembl NCBI, NGDC (National Genomics Data Center of China) there are already millions of fully sequenced genomes, most of them prokaryotic genomes. The representation of DNA sequence data is important for Artificial Intelligence tools. Analyzing such voluminous data using modern methods allows us to consider fundamental problems such as the origin of life, the principles of organization of living matter, and general patterns of transmission of genetic information during evolution. The coronavirus pandemic has sparked interest in analyzing the coronavirus genome and its subspecies (Akbari Rokn Abadi et al., 2023; Mitic et al., 2024).

The search for biologically relevant information begins with statistical approaches. Statistical study of the most (and least) common words, identification of genome regions that differ in the frequency of use of nucleotides and nucleotide words, allows us to put forward new hypotheses about the functional role of fragments of the genetic texts. Mathematical methods for estimating text complexity have been implemented in computer programs for calculating entropy, text compression complexity, and linguistic complexity of nucleotide sequences (Orlov and Potapov, 2004). It was shown the decrease in the complexity of the text and increase the variability of characters in the flanking sequences surrounding the polymorphism points (Safronova et al., 2016). The presence of single nucleotide polytracks, repeating sequences, and short tandem repeats reduces the complexity of the text. Such sites are associated with an increased frequency of mutations.

We used LZcomposer complexity assessment program (Orlov and Potapov, 2004) and own software solutions. Profiling of 10 viral genomes taken tools open NCBI databases has been performed. The research methodology included the use of web resources: REPuter, Dotmatcher, EMBOSS package, Sequence Manipulation Suite (https://www.bioinformatics.org/sms2/), Tandem Repeat Finder (TRF) (Benson, 1999). Next, visualization and analysis of low-complexity text sections were performed using the Complexity and LZcomposer web resource (Orlov et al., 2006; Orlov and Potapov, 2004) (https://wwwmgs.bionet.nsc.ru/mgs/programs/lzcomposer/).

Using a computer program analyzing linguistic complexity, fragments with low complexity matching mononucleotide repeats were found in the SARS-CoV-2 genome. An analysis of the linguistic complexity of the genome revealed a particularly significant result: the region with the least complexity encodes the S protein. This has scientific interest, since such sites have lower complexity, according to which they have an increased tendency to mutation. This observation is in good agreement with the well-known biological role of S-protein, which is a key target in the development of antiviral agents. Its structure is affected by neutralizing antibodies, it serves as the main target for viral penetration inhibitors, and it is on its basis that most modern vaccines are designed.

Calculations have shown that the maximum fragment length (perfect repeat) ranges from 7 to 51 nucleotides, and it is represented by direct repeats. It can be assumed that areas of low text complexity due to the presence of one or more repeats in DNA, which is caused by evolutionarily recent mutations, are associated with the appearance of pathogenicity properties.

It has been shown that the genomes of viruses contain short sections of direct repeats. The presence of tandem repeats (repeats of text running in a row, in two or more copies), determined using the TRF program in the genomes of viruses represented by a single-stranded RNA sequence is limited, possibly due to the lack of molecular mechanisms for their occurrence by the mechanism of DNA

slippage during replication. We found that the genomes of viruses represented by single-stranded RNA do not contain extended reverse or complementary repeats (although they could be for random reasons with a low frequency). In this case, the maximum inverted (in the complementary chain) repeat was found for the African swine fever virus, represented by double-stranded DNA. Thus, the molecular mechanisms of the occurrence of duplications associated with duplications limit the appearance of maximum perfect repeats and tandem repeats in the genomes of viruses.

## ACKNOWLEDGEMENTS

The work was supported by a grant from the Potanin Foundation for Master students Teachers in 2025 (144/25).

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
