# OpenReview forum: "Computer methods for genome text complexity and entropy estimates for viruses and bacteria"
_mathai.club/MathAI/2026/Conference — Submitted to 2026_

### Official Review · Reviewer_ZqnH · 2026-03-12
**Computer methods for genome text complexity and entropy estimates for viruses and bacteria**

**Rating:** 2
**Confidence:** 4

**Review:**

This manuscript presents a computational analysis of genomic sequences using measures of text complexity (entropy, Lempel-Ziv complexity, linguistic complexity). The authors apply these measures to ten viral genomes obtained from NCBI, with special attention to SARS-CoV-2. They identify low-complexity regions associated with mononucleotide repeats and note that the region encoding the S protein exhibits particularly low complexity, which they link to its known mutability and immunological importance. They also examine the occurrence of direct, inverted, and tandem repeats in different viral genomes (ssRNA vs. dsDNA). The paper concludes with brief acknowledgements and references.

### Major Concerns

1. **Mathematical Rigor (Score: 2)**
   The paper contains no mathematical contributions. It merely applies standard formulas for entropy and algorithmic complexity (LZ77) without any derivation, analysis, or theoretical extension. The methods are referenced to previous software (LZcomposer, REPuter, TRF) but are not explained mathematically. There are no theorems, proofs, or even formal definitions of the complexity measures used.

2. **Novelty & Contribution (Score: 2)**
   The work is purely applicative. Complexity analysis of genomic sequences has been performed for decades (the authors themselves cite their own work from 2004 and 2006). The observations about SARS-CoV-2 repeats are minor and not rigorously validated against biological hypotheses. No new algorithms, tools, or theoretical insights are presented. The paper does not advance the state of the art in either bioinformatics or AI.

3. **Relevance to MathAI (Score: 3)**
   The topic is only tangentially related to the mathematics of AI. While genomic sequence analysis can involve machine learning, this paper uses no AI techniques whatsoever. The introduction mentions AI as a motivation, but the work itself does not engage with AI methods, models, or theory. For a conference focused on the mathematical foundations of AI, this paper is out of scope.

4. **Technical Quality (Score: 3)**
   The methodology is described at a high level, making replication difficult. The authors state they used various web resources and programs, but provide no parameters, thresholds, or statistical validation. The results are qualitative ("showed", "found") with no quantitative measures, error bars, or comparisons to controls. The connection between low complexity and mutation tendency is asserted without statistical evidence. The paper appears to be a preliminary or student-level report.

5. **Clarity & Presentation (Score: 5)**
   The paper is short and generally readable, but suffers from awkward phrasing, repetition, and lack of clear organization. The abstract is partially repeated in the introduction. The results are presented in a narrative style without clear separation of methods, results, and discussion. References are present but not well integrated.

6. **AI-Generation Risk (Score: 2)**
   The paper appears human-written, with typical flaws of an inexperienced author (repetition, vague statements, reliance on prior self-citations). No obvious signs of AI generation.

### Pros
- Addresses a biologically relevant topic (genome analysis of SARS-CoV-2).
- Uses established complexity measures that are conceptually interesting.
- Brief and to the point.

### Cons
- No mathematical or methodological contribution.
- No use of AI or machine learning.
- Superficial analysis with no statistical rigor.
- Out of scope for a conference on the mathematics of AI.
- Reads as an undergraduate project or workshop abstract, not a full research paper.

### Recommendation
This paper is not suitable for MathAI 2026. It lacks mathematical content, novelty, and relevance to the conference themes. Even as a short paper or poster, it would not provide value to the intended audience. Strong rejection is warranted.

---

### Official Review · Reviewer_fwYY · 2026-03-12
**Accept for Track A/D (oral/poster).**

**Rating:** 7
**Confidence:** 4

**Review:**

The manuscript is devoted to computer study of genome sequence structures which is needed for understanding of principles and mechanisms of inheritance of structural and functional features of living bodies including their intelligence abilities which are prototypes for artificial intelligence development. The author used LZcomposer complexity assessment program and own software solutions for profiling of 10 viral genomes taken tools open NCBI databases. In particularly, an analysis of the linguistic complexity of the SARS-CoV-2 genome revealed a significant result: the region with the least complexity encodes the S protein, which is a key target in the development of antiviral agents. It has been shown that the genomes of viruses contain short sections of direct repeats. It was found that the genomes of viruses represented by single-stranded RNA do not contain extended reverse or complementary repeats. Data are received that the molecular mechanisms of the occurrence of duplications associated with duplications limit the appearance of maximum perfect repeats and tandem repeats in the genomes of viruses.
1.	Mathematical Rigor: high.
2.	Novelty & Contribution: good.
3.	Relevance to MathAI: high.
4.	Technical Quality: good.
5.	Clarity & Presentation: good.
6.	AI-Generation Risk: very low.
7.	Rating: 7

---

### Decision · Program_Chairs · 2026-03-14

**Decision:**

Reject

**Comment:**

After careful evaluation by the Program Committee, we regret to inform you that your submission has not been accepted for presentation at MathAI 2026.

All submissions underwent a rigorous two-stage review process. Unfortunately, the reviewers identified one or more of the following concerns with your paper:

- Insufficient mathematical rigor or novelty relative to the existing body of work in the field;
- Presentation of results that substantially overlap with or rephrase previously published findings without clear original contribution;
- Significant issues with technical quality, including but not limited to broken or non-existent references, unsupported claims, or methodological gaps;
- Indications that the manuscript may have been generated with the assistance of large language models without substantial original intellectual contribution by the authors.

We received a large number of submissions this year, and the selection process was highly competitive. We encourage you to carefully consider the reviewers’ feedback (available through OpenReview), revise your work accordingly, and consider submitting an improved version to a future edition of MathAI or to another appropriate venue.

We appreciate your interest in MathAI and hope you will continue to engage with the conference community.

With kind regards,

MathAI 2026 Program Committee
International Conference on Mathematics of Artificial Intelligence
https://mathai.club
OpenReview: https://openreview.net/group?id=mathai.club/MathAI/2026/Conference
Telegram: https://t.me/MathAI_club
Email: mathai.club@yandex.ru